# Direct *Salmonella* injection into enteroid cells allows the study of host–pathogen interactions in the cytosol with high spatiotemporal resolution

Chantal Ernst[1], Patrick R. Andreassen[1], Gabriel H. Giger[1], Bidong D. Nguyen[1], Christoph G. Gäbelein[1], Orane Guillaume-Gentil[1], Stefan A. Fattinger[1,2], Mikael E. Sellin[2], Wolf-Dietrich Hardt[1]*, Julia A. Vorholt[1]*

**1** Institute of Microbiology, Department of Biology, ETH Zurich, Zurich, Switzerland, **2** Science for Life Laboratory, Department of Medical Biochemistry and Microbiology, Uppsala University, Uppsala, Sweden

* hardtw@ethz.ch (WDH); jvorholt@ethz.ch (JAV)

**Data Availability Statement:** Flow cytometry data have been deposited in the Zenodo repository

## Abstract

Intestinal epithelial cells (IECs) play pivotal roles in nutrient uptake and in the protection against gut microorganisms. However, certain enteric pathogens, such as *Salmonella enterica* serovar Typhimurium (*S.* Tm), can invade IECs by employing flagella and type III secretion systems (T3SSs) with cognate effector proteins and exploit IECs as a replicative niche. Detection of flagella or T3SS proteins by IECs results in rapid host cell responses, i.e., the activation of inflammasomes. Here, we introduce a single-cell manipulation technology based on fluidic force microscopy (FluidFM) that enables direct bacteria delivery into the cytosol of single IECs within a murine enteroid monolayer. This approach allows to specifically study pathogen–host cell interactions in the cytosol uncoupled from preceding events such as docking, initiation of uptake, or vacuole escape. Consistent with current understanding, we show using a live-cell inflammasome reporter that exposure of the IEC cytosol to *S.* Tm induces NAIP/NLRC4 inflammasomes via its known ligands flagellin and T3SS rod and needle. Injected *S.* Tm mutants devoid of these invasion-relevant ligands were able to grow in the cytosol of IECs despite the absence of T3SS functions, suggesting that, in the absence of NAIP/NLRC4 inflammasome activation and the ensuing cell death, no effector-mediated host cell manipulation is required to render the epithelial cytosol growth-permissive for *S.* Tm. Overall, the experimental system to introduce *S.* Tm into single enteroid cells enables investigations into the molecular basis governing host–pathogen interactions in the cytosol with high spatiotemporal resolution.

## Introduction

Invasive enteropathogens can cause severe gastrointestinal diseases. Upon ingestion of contaminated foods, the pathogens can expand within the intestinal lumen and invade intestinal epithelial cells (IECs). One such pathogen is *Salmonella enterica* serovar Typhimurium

(https://doi.org/10.5281/zenodo.10829181) All other relevant data are within the paper and its Supporting Information files.

**Funding:** This work was supported by an European Research Council Advanced Grant (SYMBIOSES, 883077) to JAV. PRA is supported by a Novo Nordisk Foundation "Postdoc Fellowship for Research Abroad 2020 - Bioscience and Basic Biomedicine" grant (NNF20OC0059485). Work in WDH's lab is supported by grants from the SNF (NCCR Microbiomes 51NF40_180575, NRP79). The funders had no role in study design, data collection and analysis, decision to publish, or preparation of the manuscript.

**Competing interests:** The authors have declared that no competing interests exist.

**Abbreviations:** AFM, atomic force microscopy; ASC, apoptosis-associated speck-like protein containing a CARD; FluidFM, fluidic force microscopy; FRFP, far-red fluorescent protein; G6P, glucose-6-phosphate; GFP, green fluorescent protein; IEC, intestinal epithelial cell; LPS, lipopolysaccharide; NAIP, NLR family apoptosis inhibitory protein; NLR, nucleotide-binding, leucine-rich repeat; NLRC4, NLR family CARD-containing protein 4; PA, protective antigen; PAMP, pathogen-associated molecular pattern; PRR, pattern recognition receptor; *S.* Tm, *Salmonella enterica* serovar Typhimurium; T3SS, type III secretion system.

(*S.* Tm). *S.* Tm is a leading cause of gastroenteritis, which may lead to life-threatening infections [1]. The gut infection involves several coordinated steps. Using flagellar motility *S.* Tm reaches IECs, where it induces its uptake by translocation of effector molecules through the type III secretion system (T3SS) encoded in SPI-1 (T3SS-1) [2]. Inside the host cell, the bacteria reside within an *S.* Tm-containing vacuole, where T3SS-1 is down-regulated and a second T3SS encoded in SPI-2 (T3SS-2) is up-regulated [3]. In epithelial cells, *S.* Tm can escape into the cytosol and has in host cell models been observed to hyper-proliferate in this niche [4,5]. During this phase, *S.* Tm expresses T3SS-1 and flagella, seemingly ready for their next invasion upon epithelial cell extrusion [4,6,7]. In fact, the cytosolic escape cycle of *S.* Tm in IECs has been suggested as a driver for intestinal expansion [8,9].

Host organisms have evolved several strategies to defend themselves against invading pathogens, one of which is cell-autonomous innate immunity [10,11]. This critical defense mechanism operates at the individual cell level and protects against a broad range of pathogens. It is mediated by pattern recognition receptors (PRRs) that recognise pathogen-associated molecular patterns (PAMPs). Cytosolic PAMPs are recognised by inflammasomes, such as the well-characterised NAIP/NLRC4 inflammasome [12]. NAIPs (nucleotide-binding, leucine-rich repeat (NLR) family apoptosis inhibitory proteins) are PRRs localised in the cytosol and bind specific PAMPs, i.e., bacterial flagellin and T3SS components of invasive pathogens such as *Salmonella* spp. [13–15]. Upon PAMP binding, NAIPs assemble into a complex with NLRC4 (NLR family CARD-containing protein 4), usually followed by association with ASC (apoptosis-associated speck-like protein containing a CARD) and pro-Caspase-1 (cysteine-dependent aspartate-specific protease 1) [12]. Following auto-proteolytic activation, Caspase-1 processes proinflammatory cytokines (e.g., IL-1β and IL-18) and cleaves the pore-forming protein Gasdermin D, leading to pyroptotic cell death and cytokine release [16–19].

The NAIP/NLRC4 inflammasome is of major importance in combating *S.* Tm infections [5,7,20–23]. It was shown that the response is particularly crucial in IECs, where it limits *S.* Tm loads in the epithelium by driving expulsion of infected IECs into the gut lumen, as well as restricts systemic dissemination of the pathogen [5,7]. In murine IECs, the NAIP/NLRC4 inflammasome components are highly expressed [7], further indicating the importance of a rapid response upon recognition of the cognate PAMPs in this cell type.

To study the dynamics of host responses to *S.* Tm infection at the cellular level, transformed/immortalised epithelial cell lines such as HeLa, Caco2, HEK293T, and m-IC$_{C12}$ have commonly been used. However, findings obtained with cell lines may be misleading because they only partially mimic intestinal tissue characteristics [24–26]. For example, the cells are often compromised in triggering programmed cell death mechanisms, including pyroptosis, due to low or undetectable expression of the inflammasome components [24], rendering them potentially problematic for studying such host responses.

The use of enteroids can close the gap between transformed/immortalised cell lines and complex *in vivo* models. Enteroids are small intestinal epithelial organoids generated from adult stem cells [27,28] and have emerged as a powerful model for studying host–pathogen interactions with high biological relevance [9,21,29–32]. Enteroids are usually maintained in 3D, recapitulating the *in vivo* architecture of the intestinal epithelium. The gene and protein expression profile of enteroids resembles that of the original tissue, including expression of NAIPs, NLRC4, and ASC [24,33]. Recent advances in culture techniques also allow the adoption of 3D-enteroids for growth on flat surfaces as epithelial monolayers [34–36], facilitating the use of high-resolution microscopy techniques.

For almost a century, scientists have been using microinjection as a tool to study intracellular mechanisms [37]. Various cargos such as liquids, microbeads, mitochondria, or sperm

have been injected into target cells, mostly oocytes. A few publications documented the injection of bacterial species, including *Listeria* and *Salmonella* spp., into mammalian cells, specifically into transformed/immortalised cells such as Caco2 and HeLa, or macrophages [38,39]. However, conventional microinjection experiments have, to our knowledge, not yet been conducted using enteroid cells and are challenging to conduct due to their inherent fragility.

An injection technology that is minimally invasive due to precise force-control of the injection tip during injection is fluidic force microscopy (FluidFM) [40]. It combines atomic force microscopy (AFM) and microfluidics to provide a platform for the manipulation of biological samples at the nanoscale [40–44]. The technology enables the injection of small volumes of liquids into single cells in the fL to pL range with subcellular resolution and preservation of cell viability [45]. Recent advancements in tip design further enabled the handling of larger objects allowing for transplantation of mitochondria from one mammalian cell to another [46], and the injection of bacteria into the cytosol of HeLa cells [47]. This provides a unique opportunity to assess host–pathogen interactions in the host cell's cytosol uncoupled from virulence factor-mediated invasion and escape into the cytosol.

In this study, we applied FluidFM to inject *S*. Tm into single enteroid cells grown as epithelial monolayers and investigate the subsequent cytosolic responses. Employing a fluorescent live-cell inflammasome reporter and real-time microscopy, we monitored inflammasome activation following injection. By placing bacteria directly into the cytosol, our approach allowed the testing of invasion-negative *S*. Tm mutants, to investigate the relative contribution of NAIP ligands to recognition, and to assess bacterial proliferation competence in a naïve cell layer. Overall, FluidFM provided access to the cytosol of IECs and enabled the study of cytosolic host–pathogen interactions, such as inflammasome activation and *S*. Tm growth and its restriction within this niche in real time.

## Results

### Accessibility of 2D-enteroids for FluidFM-mediated solution and bacteria injection

Enteroid cells have, to our knowledge, not been subjected to single-cell injections. Therefore, we tested the feasibility of FluidFM-based injections. We opted for a 2D-enteroid monolayer to allow access of the FluidFM tips from the apical side of the IECs and, concomitantly, high-resolution imaging of the manipulated single cells with an inverted confocal laser scanning microscope from the basal side (Fig 1A). To generate 2D-enteroids, intestinal crypts were isolated from C57BL/6 mice and cultured in stem cell-maintaining conditions as 3D-enteroids. These were then disrupted, seeded, and grown as flat confluent monolayers in microscopy dishes. For injection into single IECs, we used sharpened cylindrical FluidFM tips (Fig 1A), which feature an opening of about 2 $\mu m^2$ and allow liquids and bacteria to pass through [47]. The injection process, illustrated in S1 Fig, involves the positioning of the FluidFM tip above the targeted cell in x/y under brightfield microscopy monitoring. The tip is then driven down (z) into the cell using force-feedback control. Injection is initiated by applying overpressure within the microfluidic system and can be monitored in real-time with brightfield and fluorescence microscopy. After injection has been stopped by returning to atmospheric pressure, the tip is withdrawn and can be repositioned to inject another cell. Using this technical setup and 2D-enteroid culturing protocol, we demonstrated successful injection of a dye into individual IECs within the monolayer (Fig 1B).

Next, bacteria injection was tested. We targeted individual IECs with a FluidFM probe loaded with fluorescently labelled *S*. Tm and obtained phenotypic data on cell fate by monitoring the injected cell with fluorescence and brightfield microscopy. In contrast to dye injections,

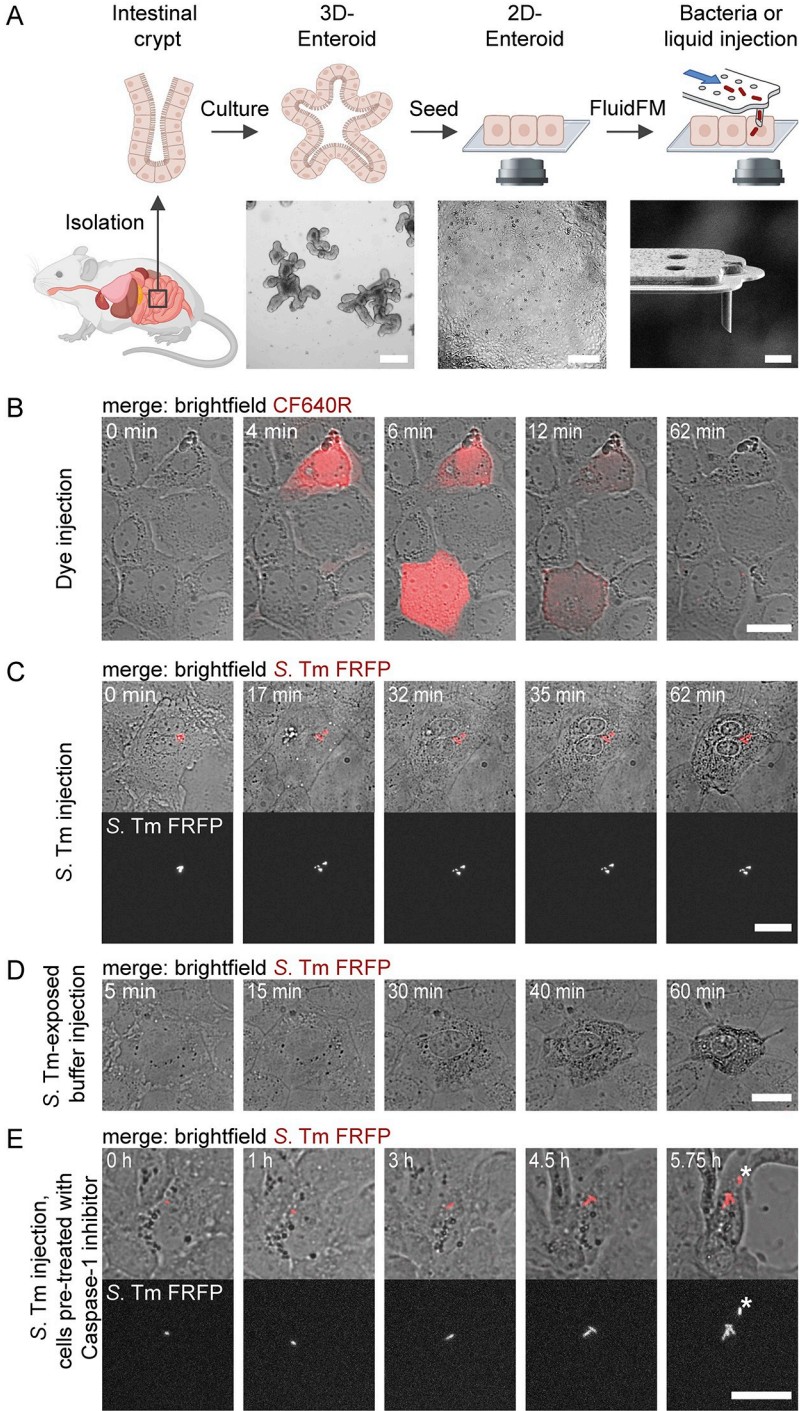

**Fig 1. FluidFM-mediated injections of liquids and bacteria into enteroid-derived IECs. (A)** Enteroid preparation for FluidFM-mediated injection. Murine intestinal crypts are isolated, embedded in Matrigel and cultured as 3D-enteroids. Subsequent seeding of enteroid IECs into a microscopy dish coated with a thin layer of Matrigel leads to the establishment of 2D-enteroid monolayers that are accessible for FluidFM manipulations. Bacteria or liquid injections are carried out using FluidFM probes with a sharpened cylindrical tip and an aperture size of approximately 2 μm$^2$. Illustration created with BioRender.com; not to scale. Brightfield images of enteroids: Scale bars: 200 μm. Focused ion beam image of a FluidFM tip: Scale bar: 5 μm. **(B)** Dye injection into single IECs. Two individual IECs were injected with a FluidFM probe loaded with injection buffer (HEPES-2) containing a far-red fluorescent tracer dye (CF640R hydrazide; f.c. 0.1 mg/ml). Fluorescence microscopy was used to monitor and verify injection into individual cells. While dye intensity decreased over time, probably due to quenching or bleaching, time lapse brightfield images show

that the cells were viable and remained within the monolayer. (**C**) *S.* Tm injection into an IEC. An individual IEC was injected using a FluidFM probe loaded with FRFP-labelled *S.* Tm in HEPES-2 buffer. While the injection of solution was monitored using brightfield microscopy (inflation of the cell), fluorescence microscopy verified successful bacteria delivery. (**D**) *S.* Tm-exposed buffer injection into an IEC. An individual IEC was injected using a FluidFM probe loaded with FRFP-labelled *S.* Tm in HEPES-2 buffer. While solution injection was monitored in brightfield, fluorescence microscopy revealed that no bacterial cells were delivered together with the buffer. (**E**) *S.* Tm injection into an IEC pre-treated with Caspase-1 inhibitor. IECs were incubated with Caspase-1 inhibitor for 1 h prior to injection. An individual IEC was injected with a FRFP-labelled *S.* Tm. Gentamicin was added to the medium 3.3 h post injection. The asterisk marks an extracellular bacterium that drifted through the field of view. Scale bars in B–E: 20 μm.

where cells sustained viability and remained within the monolayer for more than 60 min after injection (Fig 1B), we observed cell death about 30 to 60 min post bacteria injection, as judged by the change of contrast of the nucleus, followed by cell shrinkage, membrane blebbing and eventually cell expulsion from the monolayer (Fig 1C). We noted that cell death was independent of whether bacteria were delivered or whether only buffer surrounding the bacterial cells within the FluidFM probe was injected (Fig 1D), suggesting that cell death-inducing molecules might be released from the bacteria into the buffer. Because cell death of the injected cells resembled pyroptotic cell death, which is consistent with observations on IECs responding to *S.* Tm infection [20], we tested bacteria injection into IECs that were treated with a Caspase-1 inhibitor. Notably, the injected cell remained alive for up to 6 h post injection and *S.* Tm cytosolic growth was observed (Fig 1E). The addition of gentamicin, an antibiotic that is impermeable to intact mammalian cell membranes, to the medium after injection ensured that bacterial growth occurred exclusively within viable host cells.

Taken together, these results demonstrate that FluidFM-mediated injection of *S.* Tm into single enteroid-derived IECs is feasible and likely leads to Caspase-mediated cell death.

## Implementation of a real-time fluorescent reporter to monitor inflammasome activation in single IECs

Based on the initial injection experiments, *S.* Tm appeared to be recognised in the cytosol of IECs, leading to host cell responses. This manifested as cell death in immunocompetent host cells and was consistent with the expected defense against *S.* Tm involving epithelial inflammasome activation [48]. Direct placement of bacteria or bacterial components into the cytosol of host cells, opens opportunities to monitor the dynamics of cellular responses in real-time, and to separate them from other steps of the infection process, such as host cell invasion or vacuolar egress. To ensure that the observed IEC response is genuinely a regulated cellular response, we implemented a fluorescent reporter to monitor inflammasome activation. We adopted an ASC-based reporter, more specifically an ASC::GFP fusion protein (pSELECT-mASC-GFP). ASC is distributed in the cytosol until inflammasomes are activated, at which point the protein is recruited to the inflammasome, resulting in speck formation [49]. To our knowledge, the reporter has not yet been used for murine enteroid cells, and we therefore optimised the transfection protocol (**Material and methods**, S2 Fig).

To validate the reporter and inflammasome activation in IECs, we first tested the NLRC4 inflammasome-specific activator RodTox (Fig 2). RodTox is composed of the T3SS-1 rod protein of *S.* Tm, PrgJ, fused to the N-terminal domain of Lethal Factor (LF$_N$Rod) and the protective antigen (PA), which allows the cytosolic delivery of the bacterial ligand [50,51]. As anticipated, resting ASC::GFP-expressing cells showed green fluorescence throughout the whole cell, and upon extracellular addition of RodTox one or multiple intensely fluorescent specks were formed (Fig 2A). To monitor the membrane permeability of the cells, the plasma membrane impermeable DNA dye DRAQ7 was included in the medium. Speck formation was

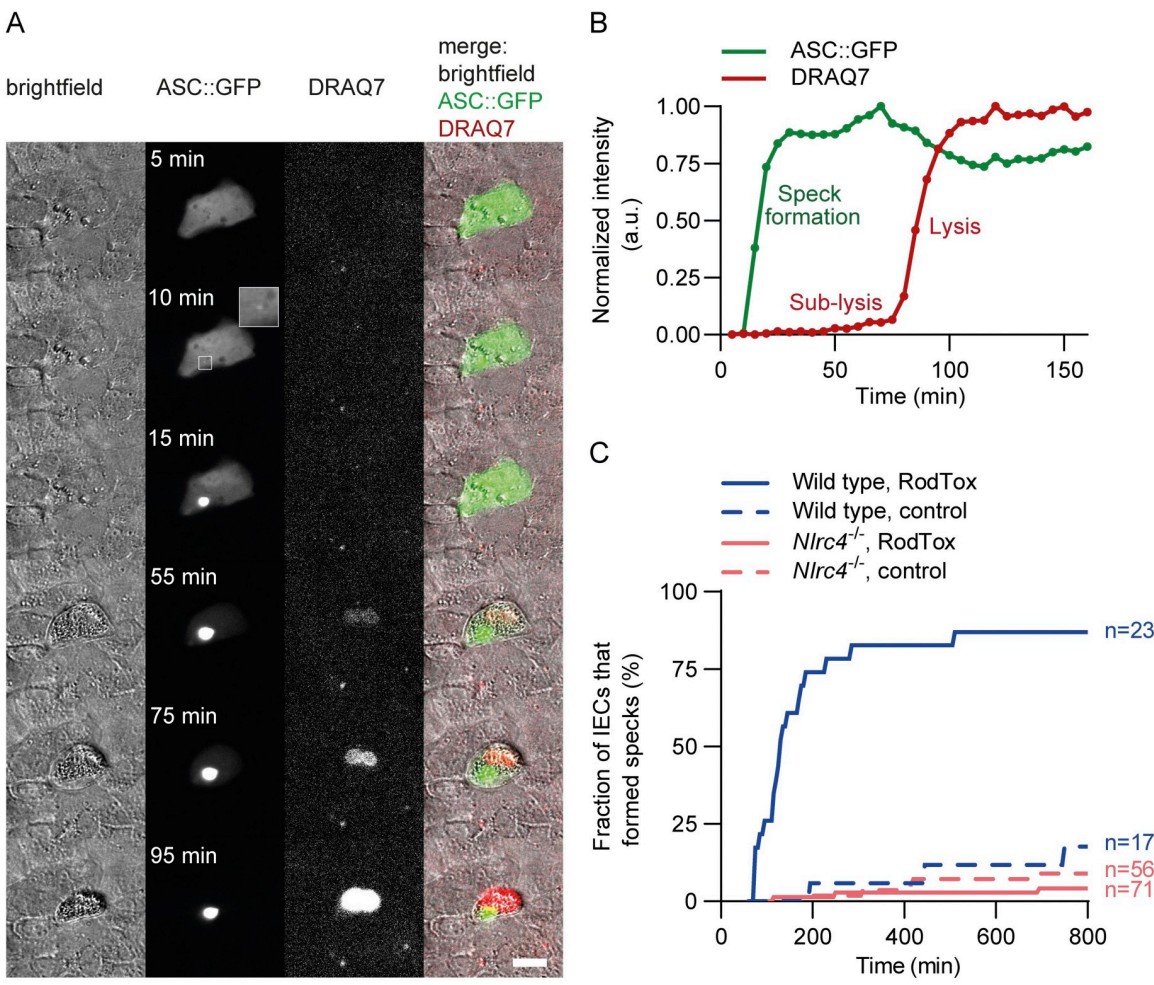

**Fig 2. Fluorescent live-cell reporter to monitor inflammasome formation in single IECs.** (**A**) Formation of a fluorescent speck in an ASC::GFP-expressing IEC upon treatment with RodTox. Cells transfected with pSELECT-mASC-GFP were treated with RodTox (LF$_N$Rod+PA). DRAQ7 was included in the medium. Scale bar: 20 μm. (**B**) Quantification of fluorescent signal at the site of speck formation (ASC::GFP) and at the cell nucleus (DRAQ7). Clear increase of the local GFP intensity indicates ASC::GFP speck formation, which is followed by a phase of slow (sub-lysis) and rapid (lysis) DRAQ7 increase. (**C**) Quantification of ASC::GFP speck formation in wild type and *Nlrc4*$^{-/-}$ IECs. The fraction of IECs reacting with speck formation upon RodTox (LF$_N$Rod+PA) and control treatment (PA only) is depicted over time. To generate comparable experimental settings to FluidFM injection experiments, these assays were conducted at room temperature. For control assays at 37°C, see S3 Fig. The data underlying Fig 2B and 2C can be found in S1 Data.

followed by slow (sub-lytic), then rapid (lytic) DRAQ7 influx, presumably due to Gasdermin D pore formation and the cell surface protein NINJ1 that drive membrane rupture [52] (Fig 2A and 2B). Concomitantly, the IECs were expelled from the monolayer (Fig 2A). Quantification of the cell fraction that reacted with speck formation revealed that this response occurs in about 80% of transfected cells, while a subfraction of cells did not readily form specks upon RodTox treatment (Figs 2C and S3). In the control treatment (PA only), only a small proportion of about 10% of transfected cells formed specks (Fig 2C), which has previously been linked to ASC::GFP overexpression [53]. When RodTox was applied to IECs lacking NLRC4 (*Nlrc4*$^{-/-}$), specks were not induced above the background level of the control treatment (Fig 2C), indicating that the induced speck formation in wild type murine enteroid cells can indeed be attributed to the NLRC4 inflammasome. In consequence, the ASC::GFP fusion

protein is a suitable single-cell reporter in enteroid-derived IECs, enabling real-time observation of inflammasome formation.

## Inflammasome assembly in single IECs in an NAIP ligand- and NLRC4-dependent manner upon FluidFM-mediated *S*. Tm solution injections

After validation of ASC::GFP as a suitable single-cell reporter for inflammasome activation in IECs, we applied the reporter to FluidFM-injection experiments using *S*. Tm. When targeting IECs with a FluidFM probe containing wild type *S*. Tm, we observed rapid ASC::GFP speck formation, on average around 10 min post targeting, in about 75% of the cells, even without delivering the bacteria themselves (Fig 3A and 3D). This observation was consistent with the strong changes in cellular morphology described above (Fig 1D), which suggested that the buffer surrounding the *S*. Tm contained sufficient amounts of cell death-inducing bacterial molecules. It is known that *S*. Tm produces and releases inflammasome-activating ligands, such as flagella and T3SS apparatus proteins, into the medium ([54,55]; S4 and S5 Figs). To test if *S*. Tm-exposed buffer is sufficient to trigger the response, we injected bacteria-free *S*. Tm supernatant obtained by centrifugation before probe loading and confirmed that the injected cells reacted with ASC::GFP speck formation (Fig 3D).

In murine IECs, the NAIP/NLRC4 inflammasome has been identified as a prominent inflammasome pathway acting upon *S*. Tm invasion [5,20]. Another inflammasome pathway involved in *S*. Tm defense, the Caspase-11 non-canonical inflammasome activated by the bacterial membrane component lipopolysaccharide (LPS), is considered to be involved only later in infection in inflamed mouse tissue [48,56]. Further, enteroids appear to require cytokine priming to sufficiently express key Caspase-11 pathway components [24,31], suggesting that the ASC specks observed upon FluidFM injections of *S*. Tm or *S*. Tm-exposed buffer are rather attributed to the NAIP/NLRC4 inflammasome. To test this hypothesis, we utilised IECs that lack NLRC4 (*Nlrc4*$^{-/-}$ IECs) or *S*. Tm that lack the specific NAIP-inducing ligands flagellin, T3SS-1 rod and needle proteins (*S*. Tm Δ*fliC*Δ*fljB*Δ*prgIJ*), while retaining the wild type LPS. Indeed, we rarely (<10%) observed ASC::GFP speck formation when testing these mutants (Fig 3B–3D). The same was true for an *S*. Tm mutant which is incapable of assembling the ligand-containing structures (flagella and the T3SS-1), and additionally the T3SS-2 (*S*. Tm Δ*invG*Δ*sseD*Δ*fliGHI*, designated as *S*. Tm Δ5) (Fig 3D). These findings confirm that ASC assembly upon cytosolic *S*. Tm solution delivery is specifically due to the formation of the NAIP/NLRC4 inflammasome and that no other bacterial compound, such as LPS, triggers ASC assembly in this experimental set up. In consequence, this enabled us to specifically study the NAIP/NLRC4 inflammasome response of IECs.

We next investigated the NAIP/NLRC4 inflammasome activation potential of various single and combinatorial *S*. Tm NAIP ligand deletion mutants (Fig 3D). We found that mutants lacking the phase one variant of flagellin FliC maintained a high potential of inducing ASC::GFP speck formation, which occurred in roughly 80% of the injected IECs. Additional deletion of *prgIJ*, encoding the rod and the needle protein of T3SS-1, barely lowered this frequency, as roughly 70% of IECs reacted with ASC assembly. This suggests that only one residual NAIP-inducing flagellin variant, here the phase two variant FljB, suffices to elicit an IEC inflammasome response to *S*. Tm solution injection. Culture supernatant analysis of these mutants showed that they indeed produced and secreted FljB at high quantities (S4 Fig). The mutant lacking both flagellin variants, *S*. Tm Δ*fliC*Δ*fljB*, had a clearly reduced rate of activating ASC::GFP specks; only roughly 40% of the IECs reacted. This suggests that the remaining NAIP ligands, PrgI and PrgJ, have the potential to activate IEC inflammasomes,

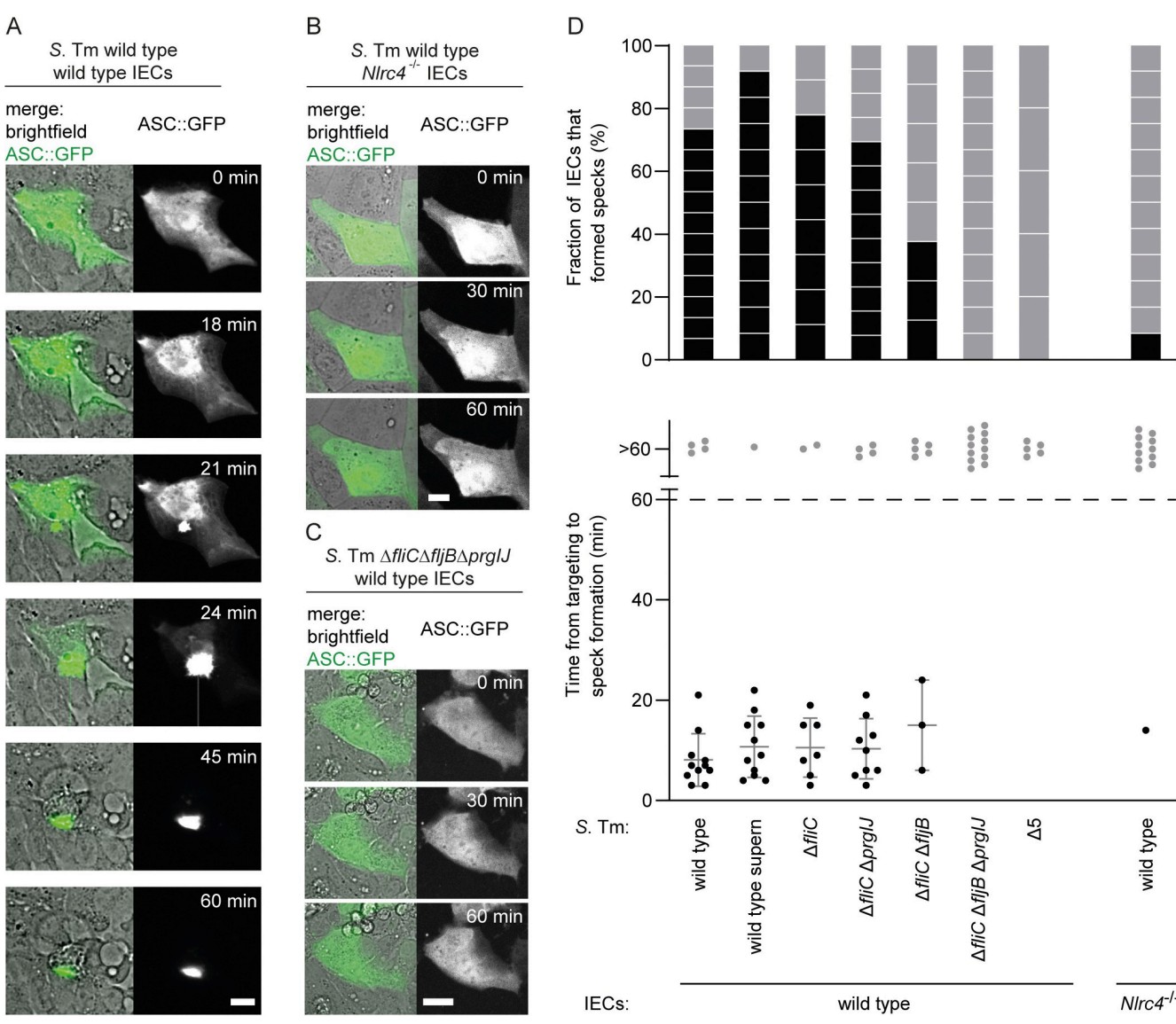

**Fig 3. Inflammasome activation in murine IECs in an NAIP ligand- and NLRC4-dependent manner upon bacterial suspension injections by FluidFM.**
(**A–C**) Time lapse images of ASC::GFP-expressing IECs injected using a FluidFM probe loaded with *S.* Tm in HEPES-2 buffer. Scale bars: 20 μm. (**A**) Wild type IEC targeted with an *S.* Tm wild type-containing probe, (**B**) *Nlrc4⁻ᐟ⁻* IEC targeted with an *S.* Tm wild type-containing probe, (**C**) wild type IEC targeted with an *S.* Tm *ΔfliCΔfljBΔprgIJ*-containing probe. (**D**) (Top) Fraction of IECs reacting with ASC::GFP speck formation within 60 min post injection with a bacteria- or bacterial supernatant-loaded FluidFM probe (black boxes) to non-reacting IECs (grey boxes). Each box represents one cell. (Bottom) Time of ASC::GFP speck formation post targeting presented as mean ± SD of the reacting cells (black dots), except for the ones with no or one measurement. Wild type IECs were targeted with FluidFM probes loaded with *S.* Tm wild type (8.1 ± 5.2 min), cell-free *S.* Tm wild type supernatant (supern; 10.7 ± 6.1 min), *S.* Tm *ΔfliC* (10.6 ± 5.9 min), *S.* Tm *ΔfliCΔprgIJ* (10.3 ± 6.0 min), *S.* Tm *ΔfliCΔfljB* (15.0 ± 9.0 min), *S.* Tm *ΔfliCΔfljBΔprgIJ*, or *S.* Tm Δ5 (*ΔinvGΔsseDΔfliGHI*). *Nlrc4⁻ᐟ⁻* IECs were targeted with FluidFM probes loaded with *S.* Tm wild type (14 min). Bacterial solutions filled into FluidFM probes had an optical density at 600 nm (OD600) of 1; the supernatant originates from a bacterial solution with an OD600 of 10 (see **Material and methods** for details). Dashed line indicates the end of monitoring (60 min post targeting), and the non-reacting cells are depicted above at >60 min (grey dots). The data underlying Fig 3D can be found in S1 Data.

but that bacteria harbouring only these ligands are less potent under the chosen conditions than bacteria that harbour a single type of flagellin.

The defined time point of cell targeting with FluidFM further enabled us to assess the dynamics of the IEC response, i.e., to quantify the timing of ASC::GFP assembly post targeting (Fig 3D, bottom). Interestingly, while the different *S.* Tm mutants varied in their induction

potential regarding the number of activated IECs, the dynamics of the cellular response did not seem to differ. If ASC::GFP assembly occurred, it robustly did so within 20 min after cell targeting.

## Cytosolic growth of invasion-deficient, NAIP ligand-lacking *S*. Tm upon injection is independent of flagella and T3SS function

The results presented above showcase that the vast majority of wild type IECs trigger the NAIP/NLRC4 inflammasome when their cytosol comes into contact with wild type *S*. Tm (Fig 3). Cells which quickly succumb to cell death do not provide an extensive cytosolic growth niche for injected bacteria. In contrast, IECs exposed to *S*. Tm mutants lacking the NAIP ligands, i.e., flagellin and T3SS-1 rod and needle proteins, did not readily activate inflammasomes (Fig 3). Since flagella and T3SSs are crucial for *S*. Tm to reach and invade IECs, studying the roles and requirements of these bacterial virulence factors in the cytosol is intrinsically challenging. FluidFM allowed us to inject these invasion-deficient *S*. Tm mutants into the cytosol and to study their cytosolic behaviour. The tagging of the bacteria with a far-red fluorescent protein (FRFP) enabled the observation of bacterial growth in real-time. We injected *S*. Tm mutants deficient in flagella and T3SS-1 function (*S*. Tm Δ*fliC*Δ*fljB*Δ*prgII*), and a mutant additionally lacking the T3SS-2 (*S*. Tm Δ5). Strikingly, both mutants were capable of proliferation within the cytosol of IECs (Fig 4A and 4B). This suggests that, in the absence of NAIP/NLRC4 inflammasome activation, T3SS function and associated effector-mediated host manipulation are dispensable for the establishment of a replicative niche for *S*. Tm in the IEC cytosol. By quantifying the fluorescent bacterial voxels over time from the fluorescence time lapse images, we recorded bacterial growth and calculated the minimal cytosolic doubling times (see **Material and methods** for details). For both strains the doubling time was 80 to 90 min at room temperature (Figs 4D, 4E and S6). To be able to compare these doubling times with cytosolic wild type growth, we also injected *S*. Tm wild type into *Nlrc4*$^{-/-}$ IECs (Fig 4C–E). Interestingly, the *S*. Tm mutants had a similar doubling time in wild type IECs as wild type *S*. Tm had in *Nlrc4*$^{-/-}$ IECs (Figs 4E and S6).

During the course of this study, we also observed two rare yet noteworthy occurrences of wild type *S*. Tm growth within wild type IECs. One of them took place under the conditions selected for bacterial growth quantification (S7A and S7B Fig). Notably, the doubling time observed was again similar to that of bacteria growing in conditions where the NAIP/NLRC4 inflammasome response is absent (Fig 4E). The second occurrence was observed during a pilot experiment conducted at 37°C where, as expected, faster cytosolic growth was observed (S7C and S7D Fig). Even though these incidences were rare, they may suggest that a subpopulation of wild type IECs fail to either detect the bacteria, or to undergo pyroptotic cell death, and thereby allow for *S*. Tm growth within their cytosol.

During natural T3SS-1-mediated infection, *S*. Tm entering the cytosol undergo transcriptional changes to respond to the new environment. One up-regulated gene is the hexose phosphate transporter *uhpT*. Gene expression driven by the P*uhpT* promoter responds to glucose-6-phosphate (G6P), a metabolite present in the host cytosol. Transcriptional fusions with fluorescent protein genes are widely adopted as valuable reporters for the cytosolic location of *S*. Tm upon invasion and vacuolar escape [3,8,57,58]. To test whether FluidFM-delivered *S*. Tm respond to the cytosolic environment in the same way as they would following natural invasion, we injected *S*. Tm harbouring P$_{uhpT}$-*gfp* into *Nlrc4*$^{-/-}$ IECs (Fig 4F). While the mean fluorescence of the constitutively expressed FRFP stayed constant in the cytosolically replicating bacteria, the mean GFP intensity of the bacterial voxels clearly increases over time (Fig 4F and 4G), indicating that P$_{uhpT}$ is induced also upon direct cytosolic delivery of *S*. Tm.

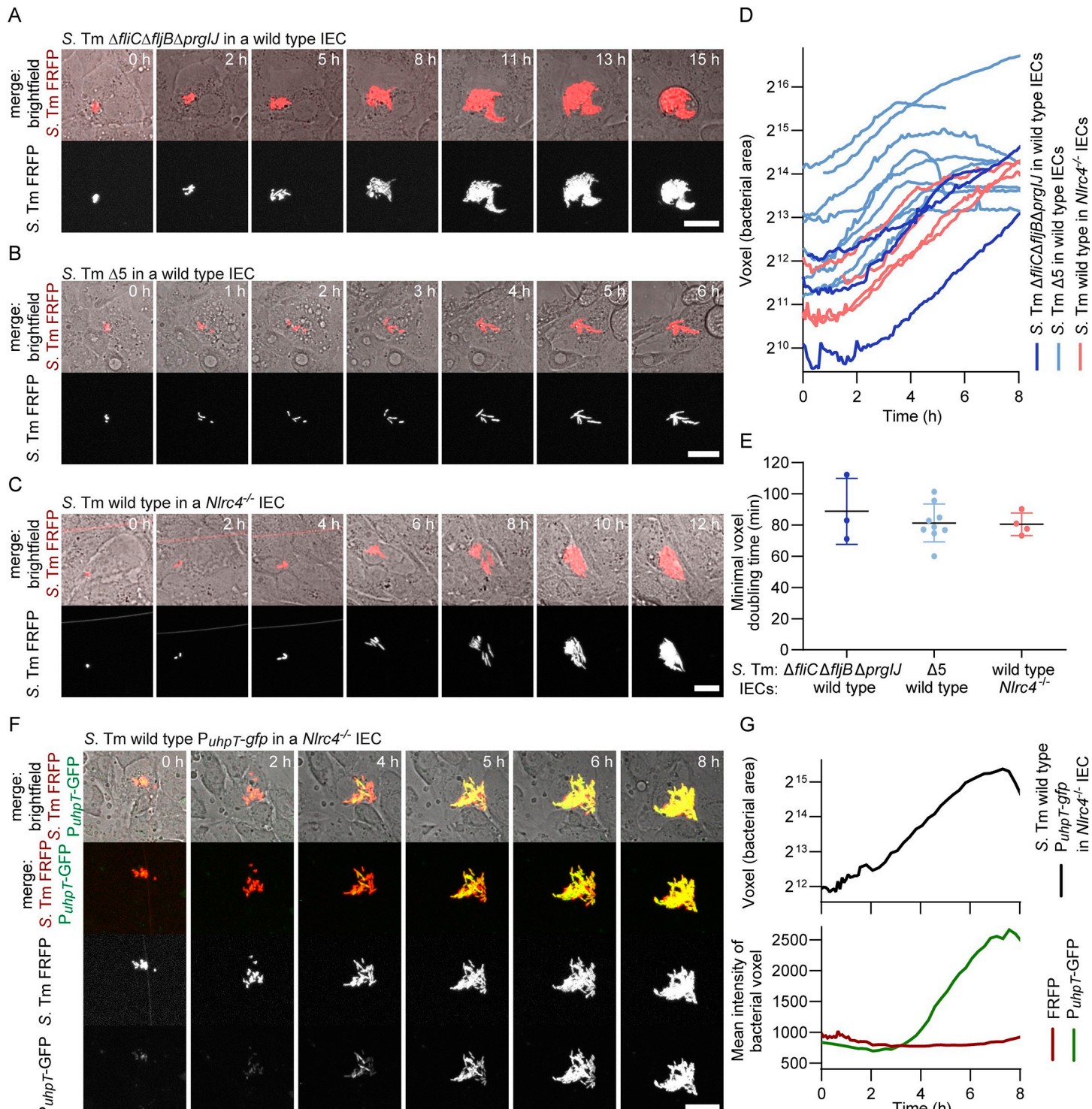

**Fig 4. Growth of *S*. Tm within the cytosol of IECs upon FluidFM injection.** (**A–C, F**) Time lapse images of individual IECs injected with FRFP-labelled *S*. Tm. Injection and subsequent imaging were done at room temperature. To ensure that the observed growth is intracellular, gentamicin was added after injection. Scale bars: 20 μm. Representative images showing (**A**) *S*. Tm Δ*fliC*Δ*fljB*Δ*prgIJ* growth within a wild type IEC, (**B**) *S*. Tm Δ5 (Δ*invG*Δ*sseD*Δ*fliGHI*) growth within a wild type IEC, and (**C**) *S*. Tm wild type growth within an *Nlrc4*⁻/⁻ IEC. (**D**) Quantification of *S*. Tm FRFP fluorescent voxels over time in IECs. Each curve shows growth of *S*. Tm in an individual IEC. (**E**) Calculated minimal intracellular voxel doubling times of *S*. Tm Δ*fliC*Δ*fljB*Δ*prgIJ* in wild type IECs (88.8 ± 21.2 min), *S*. Tm Δ5 in wild type IECs (81.3 ± 12.0 min), and *S*. Tm wild type in *Nlrc4*⁻/⁻ IECs (80.5 ± 7.2 min). Data are presented as mean ± SD. (**F**) Induction of glucose-6-phosphate-driven *gfp* expression in *S*. Tm upon injection and cytosolic growth. Representative images showing an individual *Nlrc4*⁻/⁻ IEC injected with FRFP-labelled *S*. Tm harbouring the glucose-6-phosphate-

inducible reporter P$_{uhpT}$–*gfp* (pCE052). (**G**) Quantification of *S*. Tm FRFP fluorescent voxels over time (calculated minimal intracellular voxel doubling time: 81.3 min) (top) and mean FRFP and GFP intensity of bacterial voxels over time (bottom) based on the images depicted in (F). The data underlying Fig 4D, 4E and 4G can be found in S1 Data.

In summary, FluidFM-based bacteria injection enabled the study of *S*. Tm behaviour in the cytosol of murine IECs, decoupled from invasion. Similar to bacteria reaching the cytosol via T3SS-1-mediated invasion, injected bacteria react to the cytosolic environment with the induction of the G6P-inducible promoter P$_{uhpT}$. Further, it revealed that *S*. Tm devoid of multiple virulence factors, including flagella and T3SSs, can readily proliferate within the cytosolic niche and did so as rapidly as wild type *S*. Tm in NLRC4-deficient IECs.

## Discussion

Acquiring an understanding of the cellular mechanisms by which invasive enteropathogens such as *S*. Tm induce disease and how host cells defend against infection is essential for the development of effective treatment strategies. Enteroids represent a valuable *in vivo*-like cellular model to investigate host–pathogen interactions. However, infections are typically studied in bulk assays rather than at the single-cell level, where they occur. Here, we report the applicability of the single-cell FluidFM technology to an enteroid model. This advance enabled the study of cytosolic interactions between *S*. Tm and IECs in a spatial and temporal manner with single-cell resolution.

Cytosolic *S*. Tm growth has been reported within various cultured epithelial cell lines and has been shown to considerably impact overall bacterial number, due to higher proliferation rate of the cytosolic bacteria compared to vacuolar *S*. Tm [4,59,60]. Furthermore, an *in vivo* study suggested that vacuole-escaping, cytosolic *S*. Tm in IECs influence the infection outcome by driving intestinal expansion and chronic fecal shedding [8]. We show that injection of *S*. Tm, or buffer incubated with *S*. Tm, into the cytosol of enteroid-derived murine IECs induces rapid NAIP/NLRC4 inflammasome activation and cell death (Figs 1 and 3). However, in the absence of NLRC4 or NAIP ligands, targeted IECs do not readily form inflammasomes, remain viable (Fig 3), and can represent a growth-permissive niche for cytosolically placed bacteria (Fig 4). This suggests that growth within the cytosol of IECs may be primarily restricted through NLRC4-mediated pyroptotic cell death of the host cell, likely prior to the onset of bacterial replication, while in the absence of this response, bacteria are able to exploit the cytosol as a growth-permissive environment. This is in line with previous studies, where cytosolically growing *S*. Tm are only rarely observed in NLRC4 inflammasome-competent IECs *in vivo* [7] or *in vivo*-like models such as enteroids [9,30]. In NLRC4-deficient mice however, intra-epithelial bacterial numbers are notably increased, and part of the *S*. Tm population indeed appears to be located directly in the cytosol [5]. Interestingly, we also observed rare occurrences (two throughout the whole study), where wild type IECs supported cytosolic growth of *S*. Tm (S7 Fig). Although we are currently lacking quantitative data on the fraction of cells permitting cytosolic *S*. Tm growth, the observation is interesting in context of the finding that approximately 10% to 20% of wild type IECs do not readily undergo inflammasome activation upon exposure to inducing ligands (Figs 2, 3 and S3). Together, this could indicate that there is cellular heterogeneity, or stochastic events, that render a small subpopulation of IECs incapable of triggering the inflammasome and thereby provide a growth-permissive niche for cytosolic bacteria even in an NLRC4-proficient background. While this may be the most likely explanation for these observations in our system, it is also conceivable that the pathogen might be able to suppress or evade this response, for example, via some uncharacterized T3SS effector proteins or by not displaying ligands.

While NAIP/NLRC4 inflammasomes appear to be a key mechanism in IECs to restrict cytosolic bacterial growth, the extent to which other mechanisms act, either in a parallel or in a sequential manner if inflammasome activation fails, will be interesting to explore in future research. For example, a recent study by Otten and colleagues demonstrated that *S*. Tm residing freely in the cytosol of HeLa or mouse embryonic fibroblast cells are labelled for autophagic degradation by ubiquitylation of bacterial LPS [61]. The degree to which autophagy is involved in restricting *S*. Tm growth in IECs following bacteria injection has yet to be determined. Notably, the direct placement of the bacteria into the cytosol of cells as established here is a promising approach to study autophagic processes, as it reduces the complexity of the invasion process and excludes autophagy stimuli, such as broken vacuoles occurring during phagosome escape [62].

We also investigated the NAIP/NLRC4 inflammasome activation potential of single and combinatorial *S*. Tm NAIP ligand mutants (Fig 3). While the mutants lacking all known ligands did not elicit inflammasome activation, mutants retaining one or multiple ligands activated the protein complex in a certain percentage of cells. *S*. Tm mutants lacking all ligands except for one flagellin variant (here FljB) had a high, almost wild type-like activation potential. Contrastingly, a mutant lacking flagellins but retaining the T3SS-1 proteins, PrgI and PrgJ, had a reduced capacity to induce inflammasomes. The reason for these differences in activation is currently unclear. One potential explanation could be differences in the expression levels of flagellin compared to PrgJ and PrgI under our experimental conditions, which would suggest that dosage is a determinant for NAIP/NLRC4 inflammasome activation in IECs. It has been shown *in vivo* that the controlled production of the NAIP ligands is important for *S*. Tm during infection, likely aiding immune evasion, as they promptly down-regulate flagella and T3SS-1 after IEC-traversal [7] and engineered persistent expression of flagellin leads to attenuation of the pathogen [63]. Further, we noted that the temporal dynamics of inflammasome activation was independent of the specific ligands present in the *S*. Tm mutants. If inflammasomes formed, they did so rapidly within 20 min after injection, indicating that the nature of the specific ligands plays a subordinate role and that all ligands elicit a similarly rapid response upon detection.

Attesting the relevance as targets of pathogen recognition in host cells, flagella and T3SSs are essential virulence factors for *S*. Tm to reach and infect IECs. FluidFM allowed us to inject *S*. Tm mutants that lack these invasion-relevant structures and to assess their requirement and impact for cytosolic *S*. Tm within IECs. Studies using epithelial cancer cells had revealed that the T3SS-2 is up-regulated in the vacuolar state of *S*. Tm [3], and replication in the cytosol is considered independent of the T3SS-2 encoding region SPI-2 [59,60]. In contrast, T3SS-1 and flagella are up-regulated in the cytosol [3,4,59]. Since flagella and the T3SS-1 are important for host cell targeting and cell invasion, it was proposed that the bacteria express these components to be equipped for a subsequent invasion cycle upon release from the host cell [4]. However, investigations regarding the requirement of the T3SS-1 and respective effectors for cytosolic growth may imply an additional role of these factors involved in efficient cytosol colonisation [58,59,64,65]. Here, we used enteroid-derived primary murine IECs and injected *S*. Tm deficient in these structures directly into the cytosol of these cells, thus bypassing the requirement for invasion and vacuolar stages. We show that under these conditions neither T3SS-1 nor T3SS-2 are required for bacterial growth in this niche. *S*. Tm mutants proliferated at a similar doubling rate as wild type *S*. Tm in *Nlrc4*$^{-/-}$ IECs, suggesting that, in the absence of NAIP/NLRC4 inflammasome activation, *S*. Tm does not require T3SS-1- or T3SS-2-mediated effector delivery to manipulate the host cell and render the cytosol a growth-permissive niche. However, manipulation in the sense of dampening the NAIP/NLRC4 inflammasome response in IECs would likely have a substantial impact on pathogen expansion in the gut epithelium

and subsequently at systemic body sites, as suggested by research in mice with NLRC4-deficient IECs [7]. Notably, *S*. Tm does possess T3SS effectors that affect host cell death pathways, such as SopB or SopF influencing the Akt survival pathway or Caspase-8, respectively [66,67]. However, to our knowledge, there are currently no effectors known to directly modulate the NAIP/NLRC4 inflammasome machinery in IECs. FluidFM-enabled access to the host cell cytosol could provide a valuable platform to further investigate effector-driven manipulations of host defense responses employed by *S*. Tm.

In conclusion, while impressive insights into the molecular and cellular mechanisms underlying infectious diseases have been obtained during the past decades of research, one primary challenge is the complexity of the systems involved. Here, we demonstrate how FluidFM can be used for targeted single-cell analysis of cytosolic host–pathogen interactions and dynamics in enteroid-derived epithelia.

## Material and methods

### Bacterial culture

For a list of all strains and plasmids, see S1 and S2 Tables. *S*. Tm were grown at 37°C in Luria–Bertani (Lennox) medium (LB, Sigma-Aldrich), supplemented, when needed, with carbenicillin (100 μg/ml, Huberlab AG), chloramphenicol (7.5 μg/ml, Roth), kanamycin (50 μg/ml, Roth) or streptomycin (50 μg/ml, AppliChem), or tetracycline (10 μg/ml, Sigma-Aldrich). Bacteria used for FluidFM injection experiments were fluorescently labelled with the far-red fluorescent protein TDsmURFP [68] (designated as FRFP) using the plasmid pCE047. For injection experiments, *S*. Tm strains were grown to stationary phase via overnight incubation at 37°C in 10 ml medium in 100 ml baffled flasks.

### Generation of *S*. Tm deletion strains

The *S*. Tm Δ*prgIJ* mutant was constructed using the lambda red recombination system [69], and 40-nucleotide long sequences homologous to the start of *prgI* and the end of *prgJ* were attached by PCR (oCE0157 and oCE0158, S3 Table) to the ends of the pKD4 insert containing a kanamycin resistance cassette and the Flp recognition target sites. By electroporation, the PCR products were introduced into *S*. Tm SL1344 pKD46. Recombinant bacteria were selected on LB agar plates supplemented with kanamycin. The insertion was confirmed by PCR (oCE0154 and oCE0155, S3 Table) and Sanger sequencing of the insertion with 40 nucleotides up and down-stream of the homologous sites.

To generate single and multiple knockouts in the desired *S*. Tm background (SB300), mutations were introduced and combined using P22 phage transduction. *S*. Tm Δ*fliC* was generated by introducing *fliC*::*cat* from *S*. Tm ATCC 14028 Δ*fliC* [70]. *S*. Tm Δ*fliC*Δ*fljB* and *S*. Tm Δ*fliC*Δ*prgIJ* were generated by introducing *fljB*::*kan* from *S*. Tm ATCC 14028 Δ*fljB* [70] or *prgIJ*::*kan* from *S*. Tm Δ*prgIJ* into *S*. Tm Δ*fliC*. *S*. Tm Δ*fliC*Δ*fljB*Δ*prgIJ* was generated by removing the resistance cassettes from *S*. Tm Δ*fliC*Δ*fljB* with pCP20 [71] and introducing *prgIJ*::*kan* from *S*. Tm Δ*prgIJ*. *S*. Tm Δ*invG*Δ*sseD*Δ*fliGHI* (designated as *S*. Tm Δ5) was generated by transducing *sseD*::*kan* from MvP101 [72] and *fliGHI*::Tn*10* from SB245 (*sipABCD sptP*::*kan fliGHI*::Tn*10* [K. Kaniga and J. E. Galan]) into the *S*. Tm Δ*invG* mutant SB161 [73].

### Enteroid culture

Murine enteroids were derived from C57BL/6 mice and maintained as previously described [24]. Briefly, established enteroids were maintained by embedding them in 50 μl Matrigel

(Corning) and covering the domes with Mouse IntestiCult (Stemcell Technologies) 1×Pen-Strep (Corning), incubated at 37˚C and 5% $CO_2$. Medium was replaced every 2 to 4 days and enteroids were subcultured every 5 to 7 days by mechanical sheering in Gentle Cell Dissociation Reagent (Stemcell Technologies), washing in DMEM/F-12 15 mM HEPES (Stemcell Technologies) and re-embedding in Matrigel at a 1:3 to 1:4 splitting ratio.

### Enteroid 2D-monolayer establishment and transfection

The 4-well culture inserts (Ibidi) were placed in the middle of ibiTreated low 50 mm μ-Dishes (Ibidi). The surface inside the wells were coated with Matrigel by adding 100 μl ice-cold 1:30 Matrigel:DPBS to each well and incubating at 37˚C and 5% $CO_2$ for 1 to 3 h. Matrigel-embedded 3D-enteroids grown for 3 to 4 days in Human IntestiCult (Stemcell Technologies) 1×Pen-Strep were resuspended in ice-cold DMEM/F-12 15 mM HEPES and spun at 300 rcf for 5 min at 4˚C. Cell pellets were resuspended in 1 ml 37˚C TrypLE Express (Thermo Fischer Scientific) and incubated at 37˚C in a water bath for 5 min. Enteroids were disrupted into single cells by pipetting, and 1 ml ice-cold DMEM/F-12 15 mM HEPES 1% BSA (Thermo Fischer Scientific) was added to the cells and mixed. Dissociated cells were spun at 200 rcf for 5 min at 4˚C. Cell pellets were resuspended in Human IntestiCult 1×Pen-Strep supplemented with 1 mM Y-27632 (Stemcell Technologies) and cells were enumerated with a hemocytometer. After removal of coating solution, 100 μl cell suspension with indicated cell densities were added to the Matrigel-coated wells and incubated at 37˚C and 5% $CO_2$ for 2 to 4 days. For transfection of 2D-monolayers with pSELECT-mASC-GFP (InvivoGen, psetz-mascgfp), lipofection proved efficient. Approximately 10 μl Lipofectamine 3000 (Thermo Fisher Scientific) mixture prepared according to the manufacturer's guideline was added to each well 1 to 3 h after cell seeding. Transfection efficiency was highest at low cell seeding densities; however, confluency was not achieved (S2 Fig). To strike a balance between high transfection efficiency with confluence, medium cell seeding densities (50.000 to 100.000 cells /0.35 $cm^2$) were used.

### RodTox induction of NAIP/NLRC4 inflammasomes

$LF_NRod$ (Invivogen, tlrl-rod) was reconstituted in endotoxin-free water at a concentration of 500 μg/ml and stored at −80˚C in aliquots until used. PA was expressed and purified from BL21(DE3)/pET22b-PA (Addgene) as previously described [74] with an end concentration of 300 to 500 μg/ml stored at −80˚C in aliquots until used. 2D-grown enteroids were gently washed in PBS and 100 μl $CO_2$-independent medium (Thermo Fischer Scientific) containing 10% FBS (Thermo Fischer Scientific), 1×Pen-Strep, 1×GlutaMAX (Thermo Fischer Scientific), and 1:1,000 diluted DRAQ7 (Thermo Fischer Scientific) was added to the cells. Medium was further supplemented with 5 μg/ml $LF_NRod$ and 10 μg/ml PA when indicated. Cells were incubated at room temperature (Figs 2 and S3) or at 37˚C (S3 Fig) and imaged every 5 min. The imaging set up consisted of a 20× objective on an Axiovert 200 m (Zeiss) microscope, an X/Y motorised stage (Ludl), a CSU- X1 spinning disk confocal unit (Yokogawa) (488 nm laser with ET 525/50 filter or 640 nm laser with ET 700/75 filter), and an Evolve 512 EMCCD camera (Photometrics) (Figs 2C and S3) or a 40× objective on an AxioObserver Z1 microscope (Zeiss), an X-cite light system (EXFO), fluorescence filter sets 38 HE and 50 (Zeiss), and an ORCA-Fusion BT Digital CMOS camera (Hamamatsu) (Fig 2A and 2B).

### FluidFM setup

The FluidFM system was comprised of a FlexAFM-NIR scan head controlled by the C3000 software (Nanosurf), and a digital pressure controller with a range of −800 and 1000 mbar (Cytosurge). The scan head was mounted on an inverted AxioObserver microscope (Zeiss)

that was equipped with a manual stage and a temperature-controlled incubation chamber. To generate pressure differentials greater than the range of the pressure controller, a syringe pressure kit with a three-way valve (Cytosurge) was used.

Coupled to the microscope was a spinning disk confocal microscope (Visitron) with a CSU-W1 scanner unit (Yokogawa) and an IXON Life 888 EMCCD camera (Andor). Images were acquired using a 40× objective with an additional 2× lens in brightfield and the appropriate fluorescence channels (488 nm laser, ET 525/50 filter; 640 nm laser, ET 700/75 filter) in a 16-bit format, controlled by the VisiView software (Visitron). Linear adjustment editing was made with Fiji [75]. Images in the figures were generated by using the in-focus z-slice image of the brightfield and a maximal intensity Z-projection of the fluorescence image z-stack.

## Cylindrical FluidFM cantilever processing and FIB-SEM imaging and milling

Cylindrical cantilevers with an inner cylinder diameter of 1.3 μm (Cytosurge and SmartTip) were prepared as previously described [46]. In brief, after carbon coating of the cantilever using a CCU-010 Carbon Coater (Safematic) the cylinder was sharpened by milling the tip at a 50˚ angle at an acceleration voltage of 30 kV at 40 pA by a Helios 5UX DualBeam FIB-SEM setup (Thermo Fisher) resulting in an aperture size of approximately 2 $\mu m^2$. The cantilevers were glued onto a cytoclip holder (Cytosurge). The image in Fig 1A was taken using FIB-imaging.

## FluidFM-mediated dye, bacteria, and bacterial supernatant injections

FluidFM probes were plasma cleaned (Plasma Cleaner PDG-32G, Harrick Plasma) for 1 min, vapour-coated with SL2 Sigmacote (Merck) and oven-dried at 100˚C for 1 h as previously described [42]. Probes were mounted onto the nose on the FlexAFM-NIR scan head and the cantilever spring constant was determined in air using software implemented scripts (1.5 ± 1 N/m).

For dye injection, the injection buffer HEPES-2 (10 mM HEPES, 150 mM NaCl, pH = 7.4) was mixed with the far-red fluorescent tracer dye CF640R hydrazide (final concentration 0.1 mg/ml, Sigma-Aldrich). Then, 12 μl of the dye solution were filled into the microfluidic system of the FluidFM probe.

For bacteria injection, bacterial overnight cultures were washed 3 times in HEPES-2 and adjusted to an optical density at 600 nm (OD600) of 1 to 1.5. Then, 12 μl of the bacterial solution were filled into the microfluidic system of the FluidFM probe.

For bacterial supernatant injection, bacterial overnight cultures were washed 3 times in HEPES-2 and adjusted to an OD600 of 10. The bacteria were incubated in the buffer for 1 h at room temperature. The bacterial supernatant was harvested by centrifugation, mixed with CF640R hydrazide (final concentration 0.1 mg/ml), and 12 μl were filled into the microfluidic system of the FluidFM probe.

2D-monolayers were prepared by removing the 4-well insert, washing the cells 3× with $CO_2$-independent medium containing 10% FBS to remove non-adherent cells and antibiotics and were finally kept in 5 ml of this medium. When indicated, Caspase-1 inhibitor (90 μg/ml, LabForce) was added to the medium. The cell culture dish was moved to the microscope, the scan head with a FluidFM probe was mounted, immersing the probe into the culture medium and the deflection sensitivity was calibrated at a cell-free spot on the bottom of the dish using software-implemented scripts. Then, the FluidFM tip was positioned above a target cell and the cylinder was inserted into the cell using a z-controlled force of about 50 nN. Injection was achieved by applying overpressure in the range of 5 to 50 mbar depending on the flow of

FluidFM probe. The process was observed in real-time in brightfield. After cell targeting, the pressure was reduced to atmospheric pressure and the tip was retracted. Successful delivery of fluorescently labelled bacteria or dye-containing solutions was verified using fluorescence microscopy. The scan head was removed, the lid was mounted on the culture dish to minimise evaporation and contamination, and life-cell imaging was continued to monitor the injected cells. In experiments where bacterial growth was monitored, gentamicin (final concentration 20 µg/ml, Axon Lab) was added to the medium approximately 1 to 3 h post injection to ensure that growth was truly intracellular and extracellular growth was inhibited. FluidFM injection experiments and subsequent imaging were done at room temperature to prevent temperature fluctuations during manual FluidFM operations in the incubation chamber, which can compromise the viability of the enteroid monolayers. Control experiments regarding the host response at that temperature compared to at 37˚C can be found in the Supporting information (S3 Fig).

## Bacterial growth evaluation from fluorescence time lapse microscopy images

Growth analysis of fluorescently labelled bacteria from images was done as described before [47] with some adjustments. In brief, the number of bacterial voxels per time point was calculated from the z-stack images acquired in the far-red fluorescence channel (step size 1.5 µm). To distinguish between pixels that belong to bacteria and the background pixels, a fluorescence intensity threshold of 600 was used. Subsequently, the number of bacterial voxels was plotted over time. Due to the varying growth period of bacteria in single injected cells, a minimal bacterial voxel doubling time was determined: Exponential regressions over a defined timespan of 60 min were calculated over the whole period of bacterial growth. The minimal value was taken as the minimal voxel doubling time within this specific cell and resulted in one data point in Fig 4E. See S6 Fig for graphical depiction of the doubling time calculations with the corresponding exponential fit for individual growth curves.

## Supporting information

**S1 Fig. Schematic overview of the FluidFM-mediated injection process.** The FluidFM tip is positioned (x-/y-direction) above the target cell. Next, the tip is driven down (z-direction; yellow) into the cell until the dish surface is reached, a process during which the force of the tip is monitored (red) allowing for force-feedback control of the z-movement. When inserted, overpressure is applied within the microfluidics system (blue) leading to flow of the bacterial solution into the cell. To conclude the injection, pressure is returned to atmospheric level and the tip is retracted from the cell. Created with BioRender.com.
(TIF)

**S2 Fig. Transfection rates increase with decreasing IEC density.** (**A**) Images acquired on day 3 after seeding of enteroid cells and transfecting with pSELECT-mASC-GFP, using Lipofectamine 3000. A zoomed image is depicted showing the cytosolic location of the ASC::GFP protein. (**B**) Cells at low densities are flattened with lower contrast and layers are less confluent. Dotted lines in $0.25 \times 10^5$ cells/well indicate the border of the cell layer. Scale bars: 250 µm.
(TIF)

**S3 Fig. Quantification of ASC::GFP speck formation in wild type and *Nlrc4*$^{-/-}$ IECs at room temperature (RT) and 37˚C.** Cells transfected with pSELECT-mASC-GFP were treated with RodTox (LF$_N$Rod+PA) and the fraction of IECs reacting with speck formation at RT (blue) and 37˚C (red) was quantified over time. Rep1 and rep2 indicate two biological replicas

measured in two independent experiments. The data underlying S3 Fig can be found in S1 Data.
(TIF)

**S4 Fig. SDS-PAGE analysis of *S.* Tm supernatants.** Supernatants of *S.* Tm cultures incubated overnight in LB were harvested by centrifugation, mixed with Lämmli buffer, and analysed by SDS-PAGE (mPAGE 8% Bis-Tris Precast Gel (Sigma-Aldrich), readyblue protein gel stain (Sigma-Aldrich), Precision Plus Protein Dual Color Standards (Bio-Rad)). The supernatants were loaded in an optical density-corrected manner to enable qualitative comparison of the detected protein. Culture supernatant of *S.* Tm wild type, *S.* Tm Δ*fliC*, *S.* Tm Δ*fliC*Δ*prgIJ*, *S.* Tm Δ*fliC*Δ*fljB*, *S.* Tm Δ*fliC*Δ*fljB*Δ*prgIJ*, and *S.* Tm Δ5 (Δ*invG*Δ*sseD*Δ*fliGHI*) were analysed. Protein mass of the four NAIP/NLRC4 inflammasome-inducing ligands: FliC = 51.612 kDa; FljB = 52.536 kDa; PrgJ = 10.926 kDa; PrgI = 8.857 kDa. The raw image underlying S4 Fig can be found in S1 raw images.
(TIF)

**S5 Fig. Flow cytometry analysis of T3SS-1 induction state of *S.* Tm grown *in vitro* using the T3SS-1 reporter $P_{sicA}$-*gfp* (pM972 [76]).** Bacteria were grown as for FluidFM injection experiments, i.e., to stationary phase via overnight incubation at 37˚C in 10 ml LB supplemented with appropriate antibiotics in 100 ml baffled flasks ("stationary"). Additionally, bacteria grown to late exponential phase (1:50 diluted overnight culture, 4 h, 37˚C) in LB with 0.3 M NaCl (final concentration) supplemented with appropriate antibiotics were analysed ("late exponential"). Bacteria were harvested by centrifugation, fixed in 4% paraformaldehyde (30 min, room temperature), washed 3 times in DPBS, and resuspended in DPBS. Bacteria were analysed using a flow cytometer FACSymphony A5 SE (BD Biosciences). The raw data underlying S5 Fig is available via https://doi.org/10.5281/zenodo.10829181.
(TIF)

**S6 Fig. Determination of the minimal voxel doubling time of cytosolic *S.* Tm with the corresponding exponential fit.** (**A**) *S.* Tm Δ*fliC*Δ*fljB*Δ*prgIJ* in wild type IECs (dark blue), (**B**) *S.* Tm Δ5 (Δ*invG*Δ*sseD*Δ*fliGHI*) in wild type IECs (light blue), and (**C**) *S.* Tm wild type in $Nlrc4^{-/-}$ IECs (rose). Quantification of *S.* Tm FRFP fluorescent voxels over time in IECs depicted in colour, each plot showing growth of *S.* Tm in an individual IEC. Bacterial voxel doubling times were calculated over 60 min time intervals and plotted over time (grey dots). The minimal bacterial voxel doubling time (black dot with label (min)), the respective exponential fit (black line), and the respective 60-min interval (grey filled) are indicated. The same data as in Fig 4D and 4E. The data underlying S6 Fig can be found in S1 Data.
(TIF)

**S7 Fig. Growth of *S.* Tm wild type in the cytosol of wild type IECs upon FluidFM injection.** (**A** and **C**) Time lapse images of individual IECs injected with FRFP-labelled *S.* Tm. Injection and subsequent imaging were done at (**A**) room temperature (RT) and (**C**) at 37˚C. To ensure that the observed growth is intracellular, gentamicin was added after injection. Scale bars: 20 μm. In C, the white arrow head indicates a non-intracellular bacterium that drifted away later in the time lapse and the white square indicates injected bacteria that are used for growth quantification in (D). (**B** and **D**) Quantification of *S.* Tm FRFP fluorescent voxels over time (black), showing the growth of *S.* Tm in the IEC in (A) and (C), respectively. Bacterial voxel doubling times were calculated over 60 min time intervals and plotted over time (grey dots). The minimal bacterial voxel doubling time (black dot with label (min)), the respective exponential fit (grey line), and the respective 60-min interval (grey filled) are indicated. The data

underlying S7B and S7D Fig can be found in S1 Data.
(TIF)

**S1 Data. Data underlying figures.**
(XLSX)

**S1 Raw images. Raw image underlying S4 Fig.**
(PDF)

**S1 Table. Bacterial strains used in this study.**
(PDF)

**S2 Table. Plasmids used in this study.**
(PDF)

**S3 Table. Primers used in this study.**
(PDF)

## Acknowledgments

We would like to express our gratitude to Kristin Ehrbar for constructing the *S*. Tm Δ5 mutant, Manja Barthel for providing strains and plasmids, Thea Bill Andersen for assisting with *S*. Tm wild type transformation, for providing materials to construct pCE051, and for isolating the wild type enteroids used in the study, Michael Berger for help with cell culturing, the personnel of the Flow Cytometry Core Facility (FCCF, ETHZ) for their support, and Thomas Gassler for helpful discussions.

## Author Contributions

**Conceptualization:** Chantal Ernst, Patrick R. Andreassen, Mikael E. Sellin, Wolf-Dietrich Hardt, Julia A. Vorholt.

**Formal analysis:** Chantal Ernst, Patrick R. Andreassen.

**Funding acquisition:** Patrick R. Andreassen, Wolf-Dietrich Hardt, Julia A. Vorholt.

**Investigation:** Chantal Ernst, Patrick R. Andreassen.

**Methodology:** Chantal Ernst, Patrick R. Andreassen, Bidong D. Nguyen, Christoph G. Gäbelein, Orane Guillaume-Gentil, Stefan A. Fattinger.

**Resources:** Gabriel H. Giger, Bidong D. Nguyen, Stefan A. Fattinger.

**Supervision:** Wolf-Dietrich Hardt, Julia A. Vorholt.

**Visualization:** Chantal Ernst, Patrick R. Andreassen.

**Writing – original draft:** Chantal Ernst, Patrick R. Andreassen, Julia A. Vorholt.

**Writing – review & editing:** Chantal Ernst, Patrick R. Andreassen, Gabriel H. Giger, Bidong D. Nguyen, Christoph G. Gäbelein, Orane Guillaume-Gentil, Stefan A. Fattinger, Mikael E. Sellin, Wolf-Dietrich Hardt, Julia A. Vorholt.

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
