## [Editor Report · Decision Letter 0]

9 Oct 2023

Dear Julia, 

Thank you for submitting your manuscript entitled "Single cell manipulation in enteroids – inflammasome activation upon Salmonella implantation in the epithelial cell cytosol" for consideration as a Methods and Resources Article by PLOS Biology. Please accept my apologies for the delay in getting back to you as we consulted with an academic editor about your submission. Please note that I have also provided some specific comments from the Academic Editor below my signature. 

Your manuscript has now been evaluated by the PLOS Biology editorial staff, as well as by an academic editor with relevant expertise, and I am writing to let you know that we would like to send your submission out for external peer review.

IMPORTANT: An outcome of our discussions with the Academic Editor is that we would like to consider your manuscript as a Short Report at the journal. Upon resubmission (details below), we would be grateful if you could please select 'Short Report' as the article type in the drop down menu. 

The Academic Editor has also provided some specific feedback that I have pasted below my signature (labelled 'Comments from the Academic Editor'). Given these comments, we think it would be useful to slightly reframe the paper before review and bring out the core conceptual advance in regards to the dispensability of the T3SS for the establishment of the replicative niche. To this end, we would also suggest that the sentence 'we found that targeting the cytosol of individual murine IECs with S. Tm promptly activates the NAIP/NLRC4 inflammasome' could be counterproductive to highlighting the message of the paper, since Salmonella infection is known to have this effect. For clarity, we think it would be useful to clarify that this was already known from previous work and then recapitulated when injecting S. Tm directly into the cytosol.

****

Before we can send your manuscript to reviewers, we need you to complete your submission by providing the metadata that is required for full assessment. To this end, please login to Editorial Manager where you will find the paper in the 'Submissions Needing Revisions' folder on your homepage. Please click 'Revise Submission' from the Action Links and complete all additional questions in the submission questionnaire.

Once your full submission is complete, your paper will undergo a series of checks in preparation for peer review. After your manuscript has passed the checks it will be sent out for review. To provide the metadata for your submission, please Login to Editorial Manager (https://www.editorialmanager.com/pbiology) within two working days, i.e. by Oct 11 2023 11:59PM.

Kind regards,

Richard

Richard Hodge, PhD

rhodge@plos.org

COMMENTS FROM THE ACADEMIC EDITOR

The phenomenon that Salmonella-infected epithelial cells undergo a special type of cell death has been reported by several other groups before. They had reported a dependence on the T3SS-1, presumably for two reasons: 1. since ligands of the immune system (flagelling, T3SS components) were translocated that way and 2. that the T3SS-1 is required for invasion (and possibly escape into the cytosol). The way I see is that the approach reported here is quite elegant because it bypasses the need for the invasion aspect of the T3SS-1. This is how they were able to disentangle these two T3SS-1-dependent effects, invasion vs. ligand translocation.

The conceptual advance, to me, is as follows: It was unclear whether T3SS effectors/activity were required for creating a replicative niche inside the cytosol of infected epithelial cells. Decades of research in cell culture have shown that the Salmonella T3SS-2 is required for generating a distinct replicative niche inside cells, inside a Salmonella-containing vacuole. As such, it would be logical to assume that the dependence of intracellular replication in the cytosol on the T3SS-1 could be due to effector-mediated manipulation of host cells to create a specific niche in the cytosol. The experiments reported here clearly reject this idea. This is negative data in a way, but quite useful I think.

Why did this does not fully come across: The title and abstract use overly broad and diffuse statements, which make it difficult to see the main message: "S. Tm mutants devoid of all known NAIP ligands were able to grow in the cytosol of IECs despite the absence of T3SS functions, suggesting that, in the absence of NAIP/NLRC4 inflammasome activation and the ensuing cell death, no effector-mediated host manipulation is required to render the cytosol growth permissive for S. Tm." in this model system.

---

## [Decision Letter · Decision Letter 1]

23 Nov 2023

Dear Julia,

Thank you very much for your patience while your manuscript "Salmonella injection into the cytosol of enteroid cells uncouples invasion and cytosolic host-pathogen interactions" was peer-reviewed at PLOS Biology. Please accept my sincere apologies for the delays that you have experienced during the peer review process. Your manuscript has now been evaluated by the PLOS Biology editors, an Academic Editor with relevant expertise, and by four independent reviewers. 

In light of the reviews, which you will find at the end of this email, we would like to invite you to revise the work to thoroughly address the reviewers' reports.

As you will see, the reviewers are generally very positive and agree that the study is interesting and elegant, but they do raise several overlapping concerns. Specifically, Reviewer’s #3 and #4 note that the experiments have been conducted at two different temperatures, whereby cells are maintained at 37 degrees before the injection occurs at room temperature. We ask that you please address these comments experimentally with control assays. In addition, the reviewers note that it is important to show that the injected bacteria are T3SS1-induced and that they are able to respond to the cytosolic environment in the same way as they would following T3SS1-mediated invasion.

Given the extent of revision needed, we cannot make a decision about publication until we have seen the revised manuscript and your response to the reviewers' comments. Your revised manuscript is likely to be sent for further evaluation by all or a subset of the reviewers.

**IMPORTANT - SUBMITTING YOUR REVISION**

*Re-submission Checklist*

*Published Peer Review*

*PLOS Data Policy*

*Blot and Gel Data Policy*

Best wishes,

Richard

Richard Hodge, PhD

rhodge@plos.org

REVIEWS:

Reviewer #1: This manuscript introduces the use of Fluidic Force Microscopy to inject Salmonella Typhimurium directly into the cytosol of enteroid cells. This cutting-edge technique allows host responses to cytosolic Salmonella to be examined decoupled from invasion. The authors convincingly demonstrate that delivery of STM (or even STM media) is sufficient to induce NAIP/NLRC4/ASC inflammasome activation and pyroptosis. The experiments appear carefully conducted and documented. The authors are somewhat victims of their own success and the maturity of the Salmonella pathogenesis field in that these capabilities are incremental advances over their past exemplary work and the findings are in line with expectations based on past work. Still, the authors are able to draw conclusions with greater precision in timing and in examining cytosolic delivery in isolation from other host-pathogen processes. They also are able to draw the novel conclusion that replication in the cytosol does not require flagellin or either T3SS. They also note rare occurrences of WT STM growth within WT IECs—suggesting a rare population is able to escape induction of pyroptosis that could spur further work in the future.

The manuscript is well-written. The Introduction does an admirable job recapping the state of the field and the Discussion integrates their findings well with the literature and discusses future possibilities. A couple sentences in the Introduction contrasting FluidFM with more conventional microinjection techniques would be appreciated, and perhaps an additional paragraph in the Discussion to drive home what additional questions/discoveries can now be addressed by building off the technologies highlighted in this well-executed study. One possibility is examining genetic diversity across Salmonella serovars and/or from organoids derived from different inbred or collaborative cross mice using these same tools. Or perhaps comparing to human organoids with human-specific serovars.

Reviewer #2: Summary

The authors develop an innovate way to combine microfluidics and atomic force microscopy to introduce Salmonella and bacterial products into the cytosol of intestinal epithelial cell cultures (2D organoids). They find that bacterial activation of NAIP/NLRC4 inflammasomes by flagella or T3SS rod components induces rapid cell death. Concomitantly, bacterial growth is limited in the cytosol of the cells. The paper provides an exciting new tool to study bacterial colonization of the cytosol, and the defenses that protect this compartment. The tool will be applicable to the study of other pathogens/chemical entities. I am in favour of publication with only a few comments for the authors consideration. 

Major comments

-Prior studies of bacterial growth in the cytosol of cell lines should be discussed, particularly where microinjection of bacteria was performed (PMID). 

-Figure 2A. Is the injected dye really bleaching? Or leaking out of the cell? Perhaps an external dye (e.g. PI) could be added to test this alternative possibility. 

-Figure 2A. How many cells were tested with this control? Was there no cell death ever? Since this is a new technique it is important to document the success rate. 

-Same comment for Figure 2B-D, quantification is required. Perhaps the same way data is represented in Figure 3C but with the observed phenotypes plotted over time: contrast of nucleus, cell shrinkage, blabbing, explosion, etc. Caspase inhibitor could be included. This would highlight the power of the ASC spots as an assay of cell death, since the timing seems to be very fast. 

-pg6. I appreciate that the authors include a rare occurrence where WT bacterial growth is normal within WT cells. This could be important for future studies of IECs. However, I think it is a bit jarring and misleading to include it in Figure 5D/F. I think the readers should see the true growth of WT bacteria in WT cells,.. from Figure 2B it seems there is some growth prior to cell death, which would be expected in the time scale studied. Ideally Figure 5F would have a plot of all growth rates for WT bacteria in WT cells (which would be mostly low), including the outliers that look normal. 

Minor comments

Pg4, line 152 "Rodtoxone"… add space

Pg8, line 295 "high almost wild type", a comma after "high" might make the sentence flow better

Reviewer #3: In this short report Ernst et al, have used Fluidic force microscopy to deliver Salmonella Typhimurium directly into the cytosol of cultured murine enteroid cells. This approach allows them to uncouple cytosolic events from attachment, invasion, and vacuolar escape. They then confirm that, as previously established by several laboratories, Salmonella virulence factors, specifically the T3SS1 and flagellar proteins, stimulate caspase-1 dependent cell death. Although the study does not provide any major insight into STm pathogenesis, the method described has great potential in the current "single cell" era. Overall, the paper is well written, and the experiments are technically sound, however, there are some issues that should be addressed.

Major Concerns

1. The experiments have been done at two different temperatures (37°C and room temp). The mammalian cells are initially maintained at 37°C (physiological) until the bacteria are injected. Thereafter, since the bacteria in the FluidFM system are at rt, the authors made the decision to continue at rt. However, both mammalian cells and bacterial cells behave differently at rt vs 37°C. Membrane fluidity, cell viability, motility, and growth rate can all be impacted. For example, in HL-60 cells incubation at rt has been shown to lead to caspase-1 dependent cell death (FEBS Lett. 1997 Nov 17;417(3):379-84. doi: 10.1016/s0014-5793(97)01327-6) and replication of Salmonella at room temp would likely be reduced compared to 37°C. The only experiment that was maintained at 37°C throughout, was the one in which the authors tested the requirement for caspase-1 in cell death (Fig 2D). However, given the experimental design, it possible that the observed decrease in cell death was due to the incubation temperature (37°C) rather than, or in addition to, the caspase 1 inhibitor. Appropriate controls should be included or the authors must clearly state and discuss that results from cells at rt cannot be directly compared to those from experiments done at 37°C (especially rate of bacterial replication and cell death). As presented, this (non-physiological temperature) appears to be a major limitation to the approach.

2. It would be helpful to do a direct comparison of T3SS1-mediated uptake vs FluidRM injection. Specifically, is the rate of intracytosolic bacterial replication the same in both cases? Is this dependent on temp?

3. What percentage of the STm injected into the host cells are expressing T3SS1? SPI1 expression is bistable, with SPI1-OFF and SPI1-ON populations co-existing even when STm are grown under SPI1-inducing conditions. Whereas T3SS1-mediated invasion selects for T3SS1-induced bacteria getting into the host cell this is not the case for FluidFM injected bacteria. It's important to have some information on what % of injected bacteria are T3SS1-induced and can be done using a fluorescent reporter to differentiate between SPI1-OFF and SPI1-ON bacteria both in the "inoculum" and following injection into the cytosol. I consider this essential for the study.

4. Are the bacteria injected directly into the cytosol able to respond to that environment in the same way as they would following T3SS1-mediated invasion. One way to show this would be to use a fluorescent reporter e.g. under the control of the uhpT promoter, to show whether the bacteria are respond to the cytosolic environment.

Other comments

1. The bacterial supernatant was highly concentrated compared to the bacterial injections and this should be noted in the results section.

2. Line 327- ref 61 should be included here for SopB induction of Akt

3. Line 359-361 - there are some typos.

Reviewer #4: This manuscript uses a previously developed bacterial-delivery approach to target Salmonella directly to the cytosol of host cells. The authors use this for the cytosolic delivery into enteroids, a physiologically relevant model for intestinal infection of this pathogen. Through this method, they bypass the natural cellular targeting, entry, trafficking and vacuolar escape. Therefore, they are able to test bacterial mutants that play a role in upstream processes as well as events triggered within the cellular cytosol. This is very elegant. Through this method, the authors could confirm in this pertinent model the major role of the NLRC4 inflammasome triggered through flagella and T3SS components in restricting cytosolic growth. Interestingly, the phenotype presented some leakage for bacterial growth in some cells indicating that there is cell-to-cell variability that may allow escape from growth restriction in a the cytosol of a few cells. Unfortunately, the approach does not seem to have enough throughput to obtain more robust data on this cell-cell variability. Nevertheless, there are a number of excellent novel elements in this manuscript: first, the generation of the ASC-reporter in mouse intestinal enterocytes will be an interesting tool for numerous researchers. Secondly, the clearcut analysis of the important implication of NLRC4 in Salmonella growth restriction (compared with other pathways) is an important information to drive future research. I feel that a better characterization (eg through purified flagella components), and also more controls on the amount on initially delivered pathogens (it's not clear to me how many bacteria are "injected" in each cell) would be important to make this work well rounded and interesting for a large audience- please see below. The manuscript is very carefully written and analyzed reflecting the top level research of the involved labs.

Specific points:

Cytosolic delivery of intracellular bacteria avoiding the natural entry route has been a recurrent theme of infection biologists, and this has been done for more than 20 years by micro-injection. Th

---

## [Editor Report · Decision Letter 2]

29 Feb 2024

Dear Julia,

Thank you for submitting your revised manuscript "Salmonella injection into the cytosol of enteroid cells uncouples invasion and cytosolic host-pathogen interactions" for publication as a Short Report at PLOS Biology. This revised version of your manuscript has been evaluated by the PLOS Biology editors and the Academic Editor.

Based on our Academic Editor's assessment of your revision, I am pleased to say that we are likely to accept this manuscript for publication, provided you satisfactorily address the following data and other policy-related requests that I have provided below (A-I):

(A) Since your manuscript is being considered as a Short Report, we ask that you please reduce the number of main figures from 5 to 4 upon resubmission. This could be achieved by moving the schematic diagram detailing the FluidFM manipulations in Figure 1 to Supplementary Figures?

(B) We would like to suggest a couple of alternative titles that either bring out the conceptual advance of the study or more clearly highlight the potential of the technology to study Salmonella pathogenesis. After discussions within the team, we are actually unsure which flavor of title would be best given both the research/methods aspects of your work, so I thought that it would be better to provide you with both options to consider:

“Direct Salmonella injection into the cytosol of enteroid cells reveals that establishment of the replicative niche is independent of T3SS effectors”

or

“Direct Salmonella injection into the host cell cytosol allows the study of post-invasion host-pathogen interactions with high spatiotemporal resolution”

(C) You may be aware of the PLOS Data Policy, which requires that all data be made available without restriction: http://journals.plos.org/plosbiology/s/data-availability. For more information, please also see this editorial: http://dx.doi.org/10.1371/journal.pbio.1001797

-Supplementary files (e.g., excel). Please ensure that all data files are uploaded as 'Supporting Information' and are invariably referred to (in the manuscript, figure legends, and the Description field when uploading your files) using the following format verbatim: S1 Data, S2 Data, etc. Multiple panels of a single or even several figures can be included as multiple sheets in one excel file that is saved using exactly the following convention: S1_Data.xlsx (using an underscore).

-Deposition in a publicly available repository. Please also provide the accession code or a reviewer link so that we may view your data before publication. 

Figure 3B-C, 4D, 5D-E, 5G, S2, S5A-C, S6B, S6D

(D) Please also ensure that each of the relevant figure legends in your manuscript include information on *WHERE THE UNDERLYING DATA CAN BE FOUND*, and ensure your supplemental data file/s has a legend.

(E) We require the original, uncropped and minimally adjusted images supporting all blot and gel results reported in the following Figures:

Figure S3

We will require these files before a manuscript can be accepted so please prepare and upload them now. Please carefully read our guidelines for how to prepare and upload this data: https://journals.plos.org/plosbiology/s/figures#loc-blot-and-gel-reporting-requirements

(F) For figures containing FACS data (Figure S4), please provide the FCS files and a picture showing the successive plots and gates that were applied to the FCS files to generate the figure. We ask that you please deposit this data in the FlowRepository (https://flowrepository.org/) and provide the accession number/URL of the deposition in the Data Availability Statement in the online submission form.

(G) Please note that per journal policy, we do not allow the mention of "unpublished data", "data not shown" or other references to data that is not publicly available or contained within this manuscript. Please either remove mention of these data or provide figures presenting the results and the data underlying the figure(s).

(H) Please ensure that your Data Statement in the submission system accurately describes where your data can be found and is in final format, as it will be published as written there. 

(I) Please also provide a blurb which (if accepted) will be included in our weekly and monthly Electronic Table of Contents, sent out to readers of PLOS Biology, and may be used to promote your article in social media. The blurb should be about 30-40 words long and is subject to editorial changes. It should, without exaggeration, entice people to read your manuscript. It should not be redundant with the title and should not contain acronyms or abbreviations. For examples, view our author guidelines: https://journals.plos.org/plosbiology/s/revising-your-manuscript#loc-blurb

We expect to receive your revised manuscript within two weeks. 

*Published Peer Review History*

*Press*

Kind regards,

Richard

Richard Hodge, PhD

rhodge@plos.org

PLOS

---

## [Editor Report · Decision Letter 3]

21 Mar 2024

Dear Julia,

On behalf of my colleagues and the Academic Editor, Sebastian Winter, I am pleased to say that we can accept your manuscript for publication, provided you address any remaining formatting and reporting issues. These will be detailed in an email you should receive within 2-3 business days from our colleagues in the journal operations team; no action is required from you until then. Please note that we will not be able to formally accept your manuscript and schedule it for publication until you have completed any requested changes.

PRESS

Thank you once again for choosing PLOS Biology for publication and supporting Open Access publishing. We look forward to publishing your study. 

Best wishes, 

Richard

Richard Hodge, PhD

rhodge@plos.org

PLOS
